# Research on the Regional Cooperation Innovation Network of Universities in the Guangdong–Hong Kong–Macao Great Bay Area

**Yan Wang [1] and Zhihua Liu [2],***

1 School of Government, Sun Yat-Sen University, Guangzhou 510275, China; wangy493@mail2.sysu.edu.cn
2 School of Politics and Public Adimination, South China Normal University, No. 55, West of Zhongshan Avenue, Tianhe District, Guangzhou 510631, China
* Correspondence: liuzhihua@m.scnu.edu.cn

**Abstract:** As typical innovation organizations, the structure and efficiency of cooperation among universities' innovation behaviors are important influencing factors for regional innovation sustainable development. In 2019, the Chinese government promulgated the "Outline of the Development Plan of The Guangdong, Hong Kong and Macao Great Bay Area", which directly promotes a sustainable cooperation network of universities in the Great Bay Area. This study used UCINET to visualize the cooperation network of universities in Guangdong, Hong Kong, and Macao based on the cooperation data generated by 35 universities in the Guangdong–Hong Kong–Macao Great Bay Area, jointly establishing 37 professional alliances that developed 888 cooperation ties from 2017 to 2022. The results show that the current cooperative network density of universities in the Great Bay Area is high (density = 0.746), but the cohesion trend is not significant (network centralization = 26.92%); a clear circle structure has been formed. The network exhibits a narrow shape at both ends and widens in the middle; the higher the hierarchical position of universities in the region, the more likely they are to enter the core cooperation network and establish more cooperation relationships. Universities in the marginal circles find it especially difficult to initiate cooperative relationships due to their disadvantageous position in terms of limited resources and a lack of administrative intervention. The current cooperation situation still has room for expansion.

**Keywords:** sustainable regional cooperation; innovation network; the Guangdong–Hong Kong–Macao Great Bay Area

## 1. Introduction

In 2019, the Chinese government promulgated the "Outline of the Guangdong–Hong Kong–Macao Great Bay Area Development Plan", which formally proposed to make the Guangdong–Hong Kong–Macao Great Bay Area an international science and technology innovation center with global influence. However, the existence of two political systems, three customs zones, and three legal systems within Guangdong, Hong Kong, and Macao is a natural obstacle to the formation of innovation networks among organizations, which has required a new model of cooperation in the Guangdong–Hong Kong–Macao Great Bay Area to achieve more efficient collaborative innovation. To achieve this policy goal, Guangdong, Hong Kong, and Macao have taken many initiatives in the field of innovation, especially in the field of higher education; the "Guangdong–Hong Kong–Macao University Alliance" aims to achieve efficient collaborative innovation, and a stable and sustainable collaborative innovation network has formed in the past five years. Regional sustainable cooperation networks among innovative organizations have a crucial impact on the regional innovation capacity, and although many studies have explored the effect of cooperation behavior, fewer studies have explored the structures of cooperation, especially the cooperation structures

of organizations; relationships between organizational types and cooperation strategies (complementary, proximate, and hybrid) are still insufficient.

As organizations with characteristics of both "professionalization" and "administrativeization", the cooperation behaviors of universities are unique. Although many studies have been performed on cooperation among governments, social organizations, and enterprises, there are relatively few studies focusing on the cooperation behavior of university organizations. The main difficulty in conducting such research is that the cooperative behavior of universities is multi-level and extensive, which makes it difficult to determine how the external institutional environment and internal dual institutions of universities specifically affect their cooperative behavior. This study attempted to solve this difficulty by narrowing the scope of research in space and time and clarifying the research perspective: Spatially, this paper focuses on the regional cooperation of higher education organizations, thus narrowing the spatial scope of cooperative behaviors; secondly, this paper focuses on inter-college cooperation within the formal cooperation network of the Guangdong–Hong Kong–Macao University Alliance, i.e., how universities as whole organizations cooperate with other universities in cooperation networks. Finally, in terms of time, this paper focuses on the formation and development of the cooperation network among universities in Guangdong, Hong Kong, and Macao in the five years since the establishment of the Guangdong–Hong Kong–Macao University Alliance. As an important organizational form to promote the development of regional higher education cooperation, the alliance has effectively promoted the deepening and expansion of cooperation among universities in the three regions; however, there are faults in the cooperation practice, and the progress of cooperation among different universities shows different development patterns. Therefore, this study focuses on the following questions: What are the structural characteristics of the cooperation network of universities in Guangdong, Hong Kong, and Macao? Is there any relationship between the cooperative relationship network of universities and the geographical and hierarchical positions of universities? Additionally, on this basis, we further explored how regional educational administrative agencies can govern contemporary cooperative networks of colleges and universities to further promote cooperation.

In this study, we used UCINET social network analysis software to visualize and analyze the cooperation data collected from universities in the three regions to objectively map out the structural shape of the cooperation network among universities in Guangdong, Hong Kong, and Macao and carry out specific analysis on key indicators such as density, centrality, and core–edge groups to understand the structural characteristics of the network. The data source was the cooperation data generated by 35 universities in the Guangdong–Hong Kong–Macao Great Bay Area jointly establishing 37 professional alliances, which developed 888 cooperation ties from 2017 to 2022.

## 2. Literature Review

### 2.1. Innovation Network and Organization Strategy

An innovation network is often seen as a stable conduit of the cumulative process of technological and scientific progress [1]. A growing body of research demonstrates the critical role of teamwork, clusters, and R&D networks in knowledge production [2–5]. Proximity has a key influence on the formation of innovation networks, specifically cognitive, organizational, social, institutional, and geographical proximity. Organizations with higher proximity in the above aspects are more likely to form collaborative networks, and thus, promote innovation [6]. Therefore, innovation networks are more often examined at the regional level, especially for organizations with close geographical proximity [7]. In the field of business management and industrial innovation, the main strategies of organizations in selecting partners are constructed on multiple types of complementarities [8–11]. However, in the field of higher education, complementarity is not the main causal strategy adopted by universities when selecting partners in the process of promoting inter-organizational collaborative innovation networks [12]; rather, professional proximity is more important for inter-university innovation collaboration.

## 2.2. Network Characteristics and Organization Performance

For network characteristics, innovation networks are often seen as a set of relationships, both horizontal and vertical, with other actors that are of strategic significance for the exchange partners [13–16]. Collaboration provides a context for the ongoing processes of structuration that sustain the institutional fields of the participants [17]; however, the focus is different when analyzing innovation networks from both individual and overall network perspectives. For within innovation networks, the network location of the participants Is considered an important factor that influences the innovation capacity of the actors [18]. For innovation networks overall, clusters with high performance usually have two network characteristics: strong network ties and openness to new networks [19]. In general, by studying the networks of relationships in which the target actors are embedded, we can help ourselves better understand the cooperative behavior and performance of the actors [14,20]. The governance of clusters and innovation networks is increasingly seen as an important issue, whether based on individual or holistic analytical perspectives [21]. Put differently, it is important for regional governance to develop both individual-based and holistic analyses of regional innovation networks. Compared with social organizations and enterprises, universities, as both academic organizations and public sector organizations, have higher organizational stability, which comes not only from the stability of administrative bureaucracy itself but also from the stability of professional elites in universities who continue to work in a specific professional field [7].

For the organizations, cooperation is motivated by the fact that the actors will meet again, which means that the choices made by the actors will not only affect the outcome of the current behavior but also the actors' subsequent choices; this ongoing relationship ends when one of the responders moves, changes careers, dies, or files for bankruptcy. Therefore, there is an implicit chain here that has not yet been fully explored between the past performance of actors—their position in the cooperative network—and their future performance, which is a sustainable process. This paper focuses on the first half of this implicit chain in order to theoretically and statistically prepare for better exploration of the whole chain.

## 2.3. University Cooperation as a Sustainable Regional Governance Goal

The social aspiration of the construction of the Greater Bay Area is to promote the formation and development of a regional community through deep interaction in the field of public services. Regional cooperation among Guangdong, Hong Kong, and Macao involves both the integration and docking of the two social systems and the transformation of the planned economic system into a modern market economy, as well as the breaking down of many administrative barriers [22]. In this context, universities with natural professional proximity become an important area to promote regional cooperation. However, the reason for organization cooperation is not that actors sacrifice themselves to maximize overall benefits but that actors discover the benefits of cooperation for themselves [23]. With the increasing exchanges between universities in Guangdong and Hong Kong, areas of cooperative research have been expanding. If the governments and relevant departments of the two places cooperate, the policy institutions and personnel of Guangdong and Hong Kong will certainly be able to play a better and larger role in public policy decision-making in both places [24,25]. At the same time, social service, as the third function of higher education, is prompting higher education institutions to gradually become the core kinetic force of sustainable cooperative development in the region, which is also fully reflected in the externalities of quality development of regional higher education. Some scholars have proposed the idea of establishing a "Guangdong–Hong Kong Higher Education Cooperation Pilot New Zone" based on the cooperation and exchange between higher education in Guangdong, Hong Kong, and Macao in the past three decades [26].

Current research has focused more on the city level; within the Guangdong–Hong Kong–Macao Great Bay Area, Guangzhou and Hong Kong are often seen as the core of the knowledge innovation network [27,28]. However, city-level analysis is not deep enough to

develop understanding of collaborative innovation patterns among organizations. Therefore, it is vital for regional governance to understand how universities build collaborative networks in this regional context.

## 3. Materials and Methods

### 3.1. Research Methodology and Conceptual Operationalization

Social network analysis is increasingly used in areas such as social and organizational activities; thus, it has attracted the attention of academia and industry, in which the multiplicity and complexity of inter-organizational collaborative relationships can be presented in a structured and intuitive way [29]. UCINET, as social network analysis software, provides visualization tools for various types of relational network structures so that the positions and shapes of each actor in the relational network structure are presented in a more visual form [30–32]. UCINET is often used to analyze the interactions between public, private, and non-profit organizations [33,34].

In this paper, cooperation is defined as the relationship established between universities based on a specific professional alliance goal that is based on resource exchange and formal interaction. Among them, resource operationalization is the position ranking of universities in the region to which they belong; more highly ranked universities have more resources. Interaction operationalization refers to the number of cooperative relationships established between universities, and the higher the number of formal cooperative relationships established, the more frequent the interactions between universities.

### 3.2. Data Source and Processing

This study adopted the complete enumeration method for relationship data collection; the overall research targets were member universities of the Guangdong–Hong Kong–Macao higher education alliance, and the data source was the cooperation data generated by 35 universities in the Guangdong–Hong Kong–Macao Great Bay Area. From 2017 to 2022, 35 universities jointly established 37 professional alliances affiliated with the Guangdong–Hong Kong–Macao Higher Education Alliance, which are formed by universities of their own initiative with specific professional fields; their activities must follow the relevant regulations of the Guangdong–Hong Kong–Macao University Alliance.

Three bases were used to determine the above research scope and data sources: Firstly, the member universities of the alliance basically include the universities with cooperative relationships in Guangdong, Hong Kong, and Macao, and the formed cooperative relationship network can fully represent the current situation of regional higher education cooperation among Guangdong, Hong Kong, and Macao. Secondly, the cooperative alliance based on specialties is an important organizational form of innovative cooperation among universities in Guangdong, Hong Kong, and Macao at present, and is also an important explicit indicator reflecting the structure of cooperative relationships among universities in Guangdong, Hong Kong, and Macao. Finally, this paper focuses on the structure and governance of the cooperation network among universities in Guangdong, Hong Kong, and Macao, and the use of meso-level data on inter-university professional cooperation helps to elucidate the effectiveness of promoting higher education cooperation since the implementation of the Great Bay Area construction strategy in the past five years so as to provide more feasible and effective policy suggestions for cooperation governance.

In this study, we used the complete enumeration method to collect partnership data for five years to form a case-subordination data matrix based on these relationship data: "Case" refers to the actors as the unit of analysis, i.e., universities; "subordination" refers to the events in which the actors participate together, i.e., each professional alliance. The actor coding consisted of two parts: Firstly, A, B, and C were used as the regional codes of three natural subgroups of Guangdong, Hong Kong, and Macao, respectively. Secondly, the rank coding was determined by sorting all member universities of the alliance one by one according to the QS World University Ranking 2020–2021; universities that were not ranked were coded according to the entry order in official materials of the Guangdong, Hong Kong, and Macao University Alliance, among which the 24 colleges and universities

in Guangdong region are coded as A1–A24, the 9 colleges and universities in Hong Kong are coded as B1–B9, and the 7 colleges and universities in Macao are coded as C1–C7, comprising a total of 40 actors. Five colleges and universities, A16, A18, A22, A23, and A24, had not participated in the common construction of professional alliance as of 2021; thus, 35 actors actually produced effective relationship data in order to ensure actor codes could accurately represent the rank position of actors in the region to which they belong. The codes of actors who did not participate in the construction of professional alliances were retained. The event affiliation item codes were coded as 1–37 one by one according to the time when the professional alliance was established and the order of official alliance building, finally forming the original relationship data matrix.

The original matrix was formed on the following principle: If a university (actor) was involved in a professional alliance (event), it was counted as 1; if not, it was counted as 0. For different analysis procedures within the social network analysis, the relationship data used in this paper were transposed twice: When analyzing the overall network structure and density, the original one-mode relationship data were transposed to a multi-valued co-occurrence matrix. The collaboration between a university and a college in one professional alliance can be considered one relationship line; if there were two participating professional alliances, it indicates that both universities had established two cooperative relationships, and so on. The transposed matrix contained a total of 888 collaborative relationships. When analyzing individual actors' indicators, such as point-degree centrality, core–edge, and cohesive subgroups, the multi-valued co-occurrence matrix was further converted into a binary undirected matrix.

### 3.3. Structural Characteristics of Universities in Guangdong, Hong Kong, and Macao

This study first analyzed the characteristics of the hierarchical pyramid model of universities in Guangdong, Hong Kong, and Macao, which shows distinct morphological features. These differential characteristics influence, to a certain extent, the strategic choices of the three groups in establishing cooperative relationships, as well as the differences in the demands for resources and behavioral decisions of the three groups in the operation of the relationship network.

1. Guangdong universities

The pyramid of Guangdong universities is multi-level and flat, with the following basic characteristics: the total number of structures is large; the total number of all occupants and resources in the structure is very large, so the total size of the pyramid is huge; the number of levels is large, so there are more levels in the pyramid; the difference in size is small, so the difference in occupants between adjacent levels is small, which means that the hierarchical structure of the pyramid is more balanced, and the number of occupants gradually decreases as it goes up; and the difference in resources is small, i.e., the difference of resources distributed between levels is small, i.e., the resources occupied by each level of the pyramid increase layer by layer. Due to the large volume of occupants and resources relative to Hong Kong and Macao, the occupants and related resources of each level are greater, and the hierarchical pyramid of universities in Guangdong presents a flat pyramid structure with a large volume and a more balanced structure. The number of universities in the top level is small and the demand for cooperation is high, those in the middle level are eager to break the restriction of hierarchy and the demand for cooperation is high, and the demand for cooperation and resources from outside is low at the bottom.

2. Hong Kong universities

The basic characteristics of the pyramid of Hong Kong universities are as follows: The total number of structures is moderate, with a moderate total number of all occupants and resources in the structure compared with the geographical area in which they are located; and the number of levels is small, so the levels within the pyramid are typical two-level pyramids, and there are clearly more institutions at the top than at the bottom of the pyramid, which means that the differences in scale and resources between the two levels

are very large. In terms of size and resource differences, the overall level of institutions at the top is very high; thus, the pyramid exhibits a towering feature, and the resources are also more concentrated at the top of the pyramid; on the whole, there is a considerable difference in the distribution of occupants and resources among the levels of the hierarchical pyramid of Hong Kong universities, but the difference in members within the levels is small, and the distribution of resources within the levels is relatively balanced.

3    Macao universities

The basic characteristics of the hierarchical pyramid of Macao universities are as follows: The overall structure is small, but the total numbers of occupants and resources in the structure are moderate compared with the geographical area in which they are located; the number of tiers is small, and similar to that of Hong Kong higher education institutions, the tiers within the pyramid of Macao universities are also those of a typical two-tier pyramid; in contrast to the hierarchical pyramid of Hong Kong higher education institutions, the institutions at the top are significantly smaller than those at the bottom of the pyramid; the size difference Is large, and the difference In the number of universities between the upper and lower tiers is clear; and the differences in resources are small, because the number of institutions at the top is small and the overall level is moderate. In contrast to the hierarchical pyramid of higher education institutions in Hong Kong, there are fewer institutions at the top than at the bottom of the pyramid; the difference in scale is large, and the difference in the number of higher education institutions between the top and bottom levels is obvious; and the difference in resources is small, which is due to the difference in scale, with fewer institutions at the top level and a moderate overall level. Thus, the pyramid exhibits relatively uniform two-level characteristics, and the resources are distributed more evenly between the top and bottom levels.

**4. Results**

*4.1. Overall Network Visualization and Density*

UCINET V6.645 can provide visualization for various types of relationship network structures so that the position and shape of each actor in the relationship network structure can be presented in a more intuitive form. The multi-valued co-occurrence matrix was imported into UCINET, and the relationship network structure diagram of Guangdong–Hong Kong–Macao regional university cooperation was drawn using NetDraw (Figure 1). The network structure diagram mainly consisted of actors, represented by points, and relationships between actors, represented by lines, where the position of points in the network structure diagram is related to the connectivity of actors in the establishment of neighboring relationships, and the closer the point referring to an actor is to the center of the network structure, the more connectable the structural position occupied by the actor is; the thickness of lines is a visualization of the weight of the relationships between actors in the co-occurrence matrix. The more events with which two actors are co-affiliated, the thicker the lines between them, and vice versa. By combining the location of the points in the network structure with the thickness of the lines, the output cooperative relationship network diagram could be interpreted concretely.

Based on the overall perspective of the network structure, there are rich and active neighboring relationships in the cooperation network of Guangdong, Hong Kong, and Macao universities, forming a more stable overall cooperation structure and a clear core–edge area. The actors in the middle zone are mainly universities in the middle of the hierarchical pyramid, whereas those in the peripheral zone are mostly universities with lower hierarchical positions, and their positions are more scattered. By combining the positions of the points and the thickness of the lines, it can be seen that universities in the core area have higher edge weights and thicker lines representing the relationships between the actors, because they are involved in more professional alliances, whereas the semi-core and peripheral areas have lower edge weights and line densities.

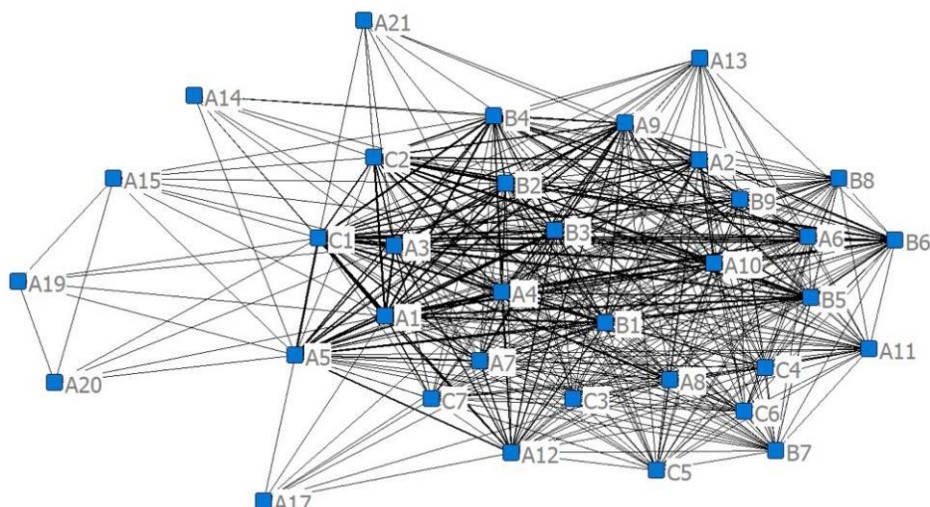

**Figure 1.** Cooperation network structure.

Network density is a concept that describes the overall level of cohesion of a network structure. The analysis of network density is based on two main metrics: the inclusiveness of the graph and the sum of the degrees of the points in the graph, where the inclusiveness of the graph refers to the total number of points contained in each associated part of the graph; the higher the inclusiveness of the graph, the higher the density. The density of a graph can be defined as the ratio of the number of connected lines that the graph actually has to the maximum number of lines that it can have. For example, for an undirected relational network containing n actors with a theoretical maximum number of relations of n (n − 1)/2, if the actual number of relations within the network is m, the density of that network is calculated as m/[n (n − 1)/2] = 2m/[n (n − 1)]. The higher the network density, the more fluid the information resources within the network and the higher the overall cohesiveness of the network. The network density analysis allows an intuitive judgement to be made about the connectivity and cohesiveness of the relationships within the collaborative network. Using the Overall Density analysis tool to analyze the overall network density, we obtained a density of 0.746 with a standard deviation of 0.435 (Table 1). This indicates a high overall network density, many active collaborative relationships within the overall network, a high level of interaction between actors, and a high level of network connectivity. The total number of lines within the network was 888 and the average number of relationship densities between actors was 25, i.e., each HEI was able to establish an average of one neighboring relationship with 25 other HEIs within the network.

**Table 1.** Overall network density.

| Density | No. of Ties | Std Dev | Avg Degree | Alpha |
|---------|-------------|---------|------------|-------|
| 0.746 | 888.00 | 0.435 | 25.371 | 0.990 |

*4.2. Individual Network Analysis*

1    Degree of connection

Point centrality refers to the number of other actors in the network that are directly related to an actor, and thus the measure of point centrality is directly based on the number of points directly connected to that point, i.e., the adjacency relationship. An adjacency relationship is a direct correlation or interconnection between the actors represented by two points, so the points adjacent to a particular point can be called the neighborhood of that point, and the total number of points in the neighborhood is called the degree of connection. Therefore, in this example of the adjacency matrix created by the relational data, the degree of a point can be directly expressed as the total number of non-zero values in the rows or columns corresponding to that point, i.e., the number of relationships established between

an actor and other actors involved in a particular professional alliance. If an actor is directly connected to more than one actor, the degree centrality of the point is high. If a point within a network has the highest degree centrality of all actors, that point resides in the center of power of that network and can transmit information resources to the greatest extent, and thus has a greater influence. Therefore, by analyzing the point-degree centrality of each point, it is possible to clarify the established neighboring relationships and influence of each actor in the network. The binarization matrix data were analyzed using the Centrality and Power-Degree analysis tool to determine the point-degree centrality of each actor within the relational network (Table 2).

**Table 2.** Degree centrality.

| Degree | Actor |
|---|---|
| 34 | A1 |
| 33 | A3, A5, C1 |
| 31 | A4, C2 |
| 30 | A9, B1, B2, B4 |
| 29 | A2, A7, A10, A12, B3 |
| 28 | A6, A8, A11, B5, B6, B9, C5, C7 |
| 27 | B7, B8, C3, C4, C6 |
| 21 | A13 |
| 10-6 | A14, A15, A17, A19, A20, A21 |
| Network centralization = 26.92%. Blau heterogeneity = 3.16%. Normalized (IQV) = 0.31% | |

The results of the analysis show that college A1, i.e., the president of the alliance, is the most influential college in the network, with a point-degree centrality of 34. C1, the vice president of the alliance, is the next most influential college, alongside A3 and A5, whereas B3, the other vice president of the alliance, is ranked next. The rank position has some influence on universities to establish neighboring relationships in the network, but there is no direct causal relationship. Universities A5, A9, A10, and B4 have established greater influence in the alliance, so the analysis of the influence of actors should be combined with the university disciplines focusing on developing and the degree of universities' own demand for cooperation with comprehensive judgment.

2    Network Centralization

Degree centralization describes the cohesiveness of a network, i.e., the degree to which the network tends to concentrate on a node. Thus, the degree centralization potential has the dual properties of overall network and individual network analysis in that the degree centralization potential focuses on the tendency of the overall network to coalesce, whereas the degree centralization potential attempts to identify the extent to which actors within the network tend to converge to a core point. Therefore, the degree centrality potential and the overall network density analysis are complementary measures, and their conclusions can support each other; the centrality potential analysis is based on the point-degree centrality analysis. First, by calculating the difference between the maximum centrality of each point and the centrality of other points, we obtained multiple "differences"; then, by calculating the sum of the differences and finally dividing the sum of the differences by the theoretical maximum possible value of the sum of the differences, we determined the degree centrality potential of the network. In this case, the degree centrality potential of the relationship network toward the core was 26.92%, i.e., the tendency of the actors in the network to coalesce toward the center of the network core was 26.92%, the cohesion trend was not significant, and the overall cohesion of the cooperative relationship network toward the core was low. When there is a natural break in the distribution of centrality or a steep drop at a specific point, it can be called the demarcation between the center and the edge; in the centrality series results, there was an obvious data break between 21 and 10, which provides an important empirical judgment basis for core–edge analysis.

*4.3. Core–Periphery Structure and Cohesive Subgroups*

1    Component Analysis

Using the Network–Region–Components–Simple graph analysis tool, the obtained network component analysis results were found to be consistent with the break in point-degree centrality, indicating that the core–edge region partitioning was reliable. The component analysis further indicated the density difference between different components within the network, and since the component analysis results are completely consistent with the partition of the core–edge, its component density can be equally regarded as the density of the core–edge region of the network. The analysis results show that the density within the core region was 0.98 and that the density was close to saturation, which indicates that the universities in the core region have established rich interactive relationships with each other: the density of the edge region was 0.2. The density of the semi-core area between the core area and the edge area was 0.24, which is similar to the density of the edge area, reflecting the unclear boundary between the semi-core area and the edge area. Therefore, cohesive subgroup analysis is needed to further supplement the results of the network core–edge analysis.

2    Cohesive subgroup analysis

Cohesion subgroup analysis focuses on the existence of sub-structures in a network, aiming to delineate the relatively strong, direct, and tightly connected subgroups of actors in an existing actor network, which have stronger cohesion among actors in the same subgroup; this cohesion is based on the relational properties between members. Wasserman and Faust proposed four perspectives for formalizing cohesion subgroups: (1) the reciprocity of relationships, (2) the proximity or accessibility between subgroup members, (3) the frequency of relationships between members within the subgroup (i.e., degree of points); and (4) the density of relationships between members within the subgroup relative to the density of relationships between internal and external members (Wasserman & Faust) [35]. Based on the analytical framework of this paper and the properties of relational data, the idea of cohesive subgroup analysis based on relational reciprocity was selected for this study. Cohesive subgroup analysis based on reciprocity mainly uses the faction (cliques) analysis procedure. The Network–Subgroup–Cliques program was run to analyze the factions within the cooperative network, and the minimum set size was established as 3, based on three natural subgroups and empirical judgments. Seven factions were obtained in the end, as detailed in Table 3.

**Table 3.** Faction structure.

| Faction | Actors |
|:---:|:---:|
| 1 | A1 C1 A3 A5 B1 B3 B4 B5 C2 A4 B2 A7 A9 B6 A10 A8 A12 A6 A2 B9 C7 A11 B7 C3 C4 C5 C6 B8 |
| 2 | A1 C1 A3 A5 B1 B3 B4 B5 C2 A4 B2 A7 A9 B6 A10 A12 A6 A2 B9 A11 C5 A13 |
| 3 | A1 C1 A3 A5 B3 B4 C2 B2 A15 |
| 4 | A1 C1 A3 A5 B4 C2 A4 B2 A9 A14 |
| 5 | A1 C1 A3 A5 A15 A19 A20 |
| 6 | A1 C1 A3 B1 C2 A4 A9 A10 A2 A21 |
| 7 | A1 A5 B1 A4 A7 A8 A12 C7 A17 |

In small-scale undirected relational networks, the composition of factions implies that the actors within a faction are more closely related (compared with members outside the faction), and the group constitutes a faction only when the members within the faction have reciprocal relationships. However, this reciprocal relationship does not constitute an exclusive relationship, and thus, there is more overlap in small-scale relational networks;

when there is more overlap in factional division, the faction overlap needs to be further analyzed. Actors who are present in more than one faction at the same time are called shared members of the group; these members have a greater capacity for reciprocal relationships in the network and thus occupy a more strategic position in the cooperative network. Based on the results of this study, some universities are present in multiple factions and are group-sharing actors in factions, among which A1 universities are present in all factions, C1 and A3 are present in six factions, and in Hong Kong, the most frequently shared universities are B1, B2, and B4, all of which are present in four factions. In this case, although Hong Kong colleges and universities have the highest ranking position among the three regions, several colleges and universities in the top ranking position in Hong Kong appear in each faction less frequently than Guangdong and Macao colleges and universities, which is closely related to the strategy of Hong Kong colleges and universities preferring to cooperate in a small number of disciplines in which they have advantages and develop cooperation needs; this is similar to institutions in the Macao region, except for C1, C2, and C4 colleges and universities. This indicates that the reciprocal relationship between Macao universities and other actors in the cooperative network is mainly limited to the single faction to which they belong, and they lack multidimensional cooperative relationships with universities in external factions.

Regarding the integration and development of universities in the three regions, faction 3 and faction 4 have realized mutually beneficial cooperative relationships among universities in the three regions, whereas faction 6 and faction 7 have relatively weak interactions, and faction 5 is dominated by Guangdong universities, indicating insufficient integration of the three regions. The main policy goal of the alliance cooperation is the integration and development of colleges and universities in three places; thus, it should further promote the formation of more balanced subgroups of cooperation among colleges and universities in the three places within the network. Secondly, it can be seen that Hong Kong and Macao colleges and universities not only have more obvious phenomena of cooperation in the analysis of group-sharing members but also in the specific analysis of each faction. Colleges and universities located in the lower position of the hierarchical pyramid of Guangdong colleges and universities can also appear in some smaller factions, but the top-heavy problem in the cooperation between Hong Kong and Macao colleges and universities is still prominent. Based on the analysis results of core–edge and cohesive subgroups, this paper divided the actors in the cooperation network of universities in Guangdong, Hong Kong, and Macao into three circles: core, middle, and edge.

*4.4. Conclusions*

In this study, we visualized the cooperative network structure of universities in Guangdong, Hong Kong, and Macao using UCINET so that the position and shape of each actor in the relational network structure could be presented more intuitively. The overall network density was analyzed, an a value of 0.746 with a standard deviation of 0.435 was obtained, which is high; the total number of inter-actor lines was 888 and the number density value was 25, which indicates that each actor was able to construct an adjacency relationship with 25 other actors within the network on average. From the analysis results of the overall cooperative relationship network, there are rich and active neighboring relationships in the three places, and a more stable cooperative network structure has been formed on the regional cooperation platform of the Guangdong–Hong Kong–Macao university alliance. The interactions between universities and the connectivity of the network are both strong and have a certain cohesiveness.

In terms of structural circles, the regional higher education cooperation network of Guangdong, Hong Kong, and Macao presents a relatively clear core–edge stratification; in this study, difficulties mainly arose due to the division between core and middle areas. Combining the analysis results of point-degree centrality and the two core–edge and cohesive subgroups, a comparative analysis of the core–middle–edge structures of the higher education cooperation networks in Guangdong, Hong Kong, and Macao was performed. The network exhibited a form of narrowing at both ends and widening in the middle. The higher the hierarchical

position of universities in the region to which they belonged, the more likely they were to enter the core of the cooperation network and establish more cooperation relationships. Universities in the marginal circles, especially, experience more difficulties in the initiation of cooperative relationships due to their disadvantageous position in resources and lack of administrative intervention. The current situation of cooperation requires further research, which is a suggested topic for future studies.

## 5. Discussion

### 5.1. Sustainable Development of Regional Higher Education Integration as a Global Trend

In the second half of the 20th century, with the global expansion and development of higher education, the higher education systems in most countries became more complex and highly differentiated, resulting in a variety of institutions. The regional imbalance of higher education has become more prominent in this process, which focuses the national, regional, and academic circles more on the regional dimension of higher education. On the one hand, in order to enhance the core competitiveness of a country in the globalized knowledge economy, the state must make policy decisions and invest in national and regional innovation systems; as the key source of knowledge and innovation, higher education institutions are the top priority. On the other hand, with the growing trend towards collaborative knowledge production, universities must do more than education and research, and must cooperate with the regions in which they are located. The relationship between higher education and regional development is being examined in a more positive and open manner by the state, the region, and universities.

Based on previous analysis of the distribution of higher education resources in Guangdong, Hong Kong, and Macao, we know that the hierarchical structural characteristics of higher education resources in the three regions exhibit obvious differences. Hong Kong and Macao are typical two-level pyramidal structures, whereas Guangdong has a typical multi-hierarchical structure. By combining the results of the hierarchical structure analysis with the network structure analysis, we can see that without the intervention of external forces, most of the deep cooperative relationships are established between universities that are at the top of the pyramid of the three group levels; structural limitations are obvious.

In the theory of organizational cooperative relationships, in a hierarchical structure, the relative sizes of groups tend to decrease gradually in an upward direction, and when the relative size gap between two levels keeps increasing, the upward inter-group interactions initiated from the lower level to the higher level decrease; however, when the relative size gap between levels decreases, the upward interactions increase [20]. For universities, this means that when there are more universities at the bottom of the pyramid and fewer universities at the top of the pyramid, fewer cooperative actions can be initiated upward by universities at the bottom. On the one hand, this is due to hierarchical restrictions, i.e., the cooperation demands of actors at the top of the pyramid can be mostly obtained through intra-tiered relationships; on the other hand, the larger the number of actors at the bottom, the smaller the chance that a particular actor can exchange resources with actors at the top. This limitation is also valid in cross-group cooperation. In this study, when cross-group cooperation was established, the actors at the top of the flat pyramid structure of Guangdong universities were found to be less likely to select actors at the bottom of the towering pyramid structure of Hong Kong universities when choosing partners across groups, because there is already a large number of actors at the top; hierarchical restrictions enabled both parties to collaborate within the layers.

The heterogeneity of cooperation demands of universities in the three regions leads to the paradox of high transaction costs, low cooperation benefits, and high cooperation incentives in regional higher education cooperation among Guangdong, Hong Kong, and Macao. Existing studies generally point out that the characteristics of goods and services provided by cooperation are an important influencing factor for cooperation decisions [36,37]. Therefore, the likelihood of cooperation depends on the judgment of cooperative agents on transaction costs and expected benefits on the one hand; on the

other hand, the difficulty of measuring and monitoring the benefits of cooperation can negatively affect the likelihood and durability of cooperation [38]. In this context, strategic decisions made before establishing a cooperative relationship are particularly critical. The findings of this study also point out that, in the absence of greater administrative intervention, it is in the universities' own interest to collaborate with institutions in higher hierarchical positions and with more resources, which is the current collaboration strategy of many universities. However, this may not match the overall aspirations of regional higher education governance; therefore, further exploration as to how the development of networks of collaborative relationships can be sustainable through hierarchical governance is vital.

*5.2. Hierarchical Governance*

Universities in the core circle are in an advantageous position in terms of both their resources and location, and they occupy strategic pathways in the cooperation network. These institutions can obtain and transmit more information and exert greater authority, which means that when new cooperation opportunities arise in the cooperation network, universities in the core circle are the most favorable competitors. Therefore, the cooperative governance of the core circle should focus on promoting fellow universities to actively exert their own advantages and play driving roles in the development of regional higher education to avoid an excessive concentration of resources. The development of regional higher education integration cannot be achieved overnight; it is a gradual exploration and gradual policy process. It is also necessary for universities in different circles and at different development levels to gradually realize integration in stages and with planning.

Universities in the middle circle usually have rich resources in specific professional fields and can compete to form alliances in their own specialist areas and compete for a structural position in the pyramid due to interaction with universities in the core circle, establishing strong foundations to expand cooperation in other fields. Therefore, universities in the middle circle should be encouraged to diversify their development and concentrate on their own specialist subject areas in regional higher education cooperation and actively establish extensive cooperation with other universities in professional fields with good cooperation bases. On the one hand, this can help universities maximize the utilization of their own resources, i.e., through cooperation in specific professional fields within a certain period. The professional influence of universities in this field can be enhanced, and good foundations can be laid for the subsequent further expansion of cooperation. On the other hand, universities in the middle circle should be encouraged to actively enhance their own advantageous disciplines and diversify development to avoid homogeneous competition in the overall regional cooperation network and promote real cluster progress.

Universities in the edge circle are in a relatively disadvantageous position; it is difficult for these institutions to initiate or become the founding universities of professional alliances because of limited resources and a limited cooperation base. Network density in the edge circle is also low, which indicates that cooperative relationships among universities here are not active and the cost of constructing cooperative relationships is high. Although universities in the marginal circle are at a disadvantage in terms of resources, regional higher education integration should focus on overall development and should start from location resources, allocating more policies and resources to universities in the marginal circle and promoting universities in the marginal circle to actively integrate into the cooperative network by combining internal and external factors. Such policies should encourage core- and middle-sphere universities to build cooperative relationship with edge-sphere universities to reduce the cost of cooperation of edge-sphere universities. The problem of the low efficiency of resource inputs and outputs can be solved by policy support, and cooperation practices can be instilled in faculties and even at individual levels to weaken the hierarchical restrictions and avoid the formal margins of cooperation. Moreover, the incentive of resources to the edge circle should be strengthened through institutional design to enhance the enthusiasm of universities in the edge circle to participate in cooperation networks inside and outside of their circle.

**Author Contributions:** Conceptualization and methodology, Y.W.; formal analysis and resources, Z.L.; writing—original draft preparation, Y.W.; writing—review and editing, Z.L. All authors have read and agreed to the published version of the manuscript.

**Funding:** This research received no external funding.

**Institutional Review Board Statement:** Not applicable.

**Informed Consent Statement:** Not applicable.

**Data Availability Statement:** The data presented in this study are available on request from the corresponding author.

**Conflicts of Interest:** The authors declare no conflict of interest.

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
