# Peer review of "Research on the Regional Cooperation Innovation Network of Universities in the Guangdong–Hong Kong–Macao Great Bay Area"

_sustainability, doi:10.3390/su15129838_

Round 1

Reviewer 1 Report

The study is well planned. The base is adequate, the methodological design successful. The conclusions respond to the proposed objectives and with current references. Its publication is recommended given the aforementioned conditions and the interest of the topic. For all the above, I consider that this work can be published.    

Author Response

Point 1:  Reviewer 1 mentioned that “the arguments and discussion of findings coherent, balanced and compelling” can be improved.

Response 1: Thank you very much for your affirmation of this article and for pointing out the parts I can improve. As you mentioned, I have further improved the presentation and discussion of the results, especially the structure of the conclusion and discussion sections, as well as the rest of the article to make it more coherent and logical. Thanks again for your valuable suggestions!

Reviewer 2 Report

What is the research question for the study?

What are the objectives of the study? Please state them clearly.

The methods section has no references. I would like to know what you based the choice of methodology on and what other studies (published in major Scopus-indexed journals) have used the same approach.

I would also like to see a conclusion section for the article. 

Author Response

Point 1:  What is the research question for the study?

Response 1: Thank you for your suggestion. In the previous version, I did not state it clearly, and during the revision process, I further clarified the research questions of this paper and revised and stated them in the Introduction section.

Point 2: What are the objectives of the study? Please state them clearly.

Response 2: In relation to point 1, I have provided a clearer statement of both my research question and my research objectives in the introduction.

Point 3: The methods section has no references. I would like to know what you based the choice of methodology on and what other studies (published in major Scopus-indexed journals) have used the same approach.

Response 3: Thank you for your question. I have strengthened my rationale for choosing this approach in the Introduction and Methods sections, respectively, by further discussing how UCINET can help us better understand inter-organizational cooperation, and as you suggested, I have added literature exploring inter-organizational cooperation that has adopted this research approach.

Point 4: I would also like to see a conclusion section for the article. 

Response 4: This is a very critical issue, and I'm sorry I didn't get it right in the previous version. In the revised version, I have specified the conclusions of this paper in "4.1 Conclusion" so that it can be better connected to the discussion section.

Thank you for your valuable suggestions, I have thoroughly considered the issues you raised and made a lot of changes to the whole text. Thank you for giving me the valuable opportunity to revise.

Reviewer 3 Report

·        Thank you for the opportunity to review the manuscript entitled “Sustainable Regional Cooperation Network: The Research on the Cooperation Innovation Network of Universities in The Guangdong-Hong Kong-Macao Great Bay Area”, by Yan Wang, Zhihua Liu, submitted for publication in Sustainability, manuscript ID sustainability-2406482.

·        Even though the topic of the paper is very interesting – University Cooperation Innovation Networks, it is hard to argue in favour of its suitability with the topic of the journal. Moreover, this version of the article needs more development in the different sections and it does not meet the necessary criteria to recommend its publication in the submitted form.

·        Below are highlighted several key points related to the manuscript and recommendations for revisions:

-        The term „sustainable” appears just in the title, in the abstract (4 times), in the keywords and once in the introduction. It appears that the article has been adapted in order to be sent to this journal, by introducing here and there the key concept of the journal, but without internalizing it in the structure of the paper.

-        The aim of the paper is not clear. It is not stated in the introduction, in the methodology, etc., being vaguely mentioned in the abstract. In general, the paper is descriptive, lacking the analytical/ data interpretation sections. There is no hypothesis or a mere research question. There is not a link between the main findings to prior literature or a highlight of the study’s contribution to the existing body of research.

-        There are some English/coherence aspects that need to be reconsidered (for example, the logic of the lines 33-34).

-        The full stop should be placed after the citation bracket, not before it.  

-        Line 58 advances the idea to use IEs for innovation ecosystems, but it is not the case for the rest of the paper, as the collocation does not appear anymore.

-        The Literature review section is not clear, unstructured, mixing general information about innovation networks ant the study cases dealing with higher education examples.

-        Materials and method – the main concepts used and not operationalized; therefore, we cannot understand what is meant by cooperative relationships and what kind of items it covers (common programmes, transfer of disciplines, collaborative teaching, and so on and so forth, as there are countless possibilities to understand this cooperative term).

-        The processed data are simply described and little is offered to understand why the networks have a peculiar aspect instead of another.

-        Discussion: „It is well known that universities, especially research universities, are relatively more concerned with knowledge innovation and development that have an impact on national and global economies” – this statement should be reconsidered, as the situation is not so well known or generally available, as there are many types of universities and ways to define the environment they belong to and operate within.

-        Lines 372 – 400: (1) It is appropriate to indicate that several cooperation mechanisms exist in other parts of the world, but this comparative analysis is not further developed in the manuscript, in terms of implications, etc. (2) it is not recommendable to compare such different cooperative examples such as the EU, AHEA or the US system, especially in connection with the aim of the current paper; (3) there are considerable errors in the EU dedicated section, as the EU (or its higher education cooperative mechanisms) is not connected to the 1948 Hague declaration, the Bologna Process was not issued by the EU (which is just one of its members), and the term European Community, as a legal body, does not exist anymore after the entering into force of the Lisbon Treaty, back in 2009. Also, we should mention that at the EU level, education does not represent a common policy, and the cooperation in several sectors (like higher education) is based on the voluntary agreements of the universities and, sometimes, of the states.

-        The last sub-section: there is no critically discussion of the results against the literature and the research questions (as they are missing). It is not clear what are the main contribution of the study and how could they inspire future research. There are several recommendations pointing towards a redistribution of resources, but, without operationalizing the type of resources envisaged, or how this recommendation was reached, the section remains unclear, as well as the very last phrase of the article „The development of resource-interaction-emotion in the marginal circle should enter a virtuous cycle”.

·        I hope my comments could help the authors improve their studies.

No comments.

Reviewer 4 Report

The study is both comprehensive and relevant. It is based on a sound research approach and enjoys a well-structured and well-written format. The arguments are sensible. However, the paper can be improved with more references and linkage with the literature. Sixteen references does not appear to sufficiently cover the issue at hand and the variety of references cited can be improved. It is suggested that the authors consider the potential of adding 5-8 more references and including these in the literature review and discussion sections of the paper. Though this weakness does not preclude the paper from being published, it would certainly enhance the paper.

Author Response

Point 1:  The paper can be improved with more references and linkage with the literature. Sixteen references does not appear to sufficiently cover the issue at hand and the variety of references cited can be improved. It is suggested that the authors consider the potential of adding 5-8 more references and including these in the literature review and discussion sections of the paper.

Response 1: Thank you very much for your suggestion, as you pointed out, in the previous version my number of references and the dialog with the references were insufficient. Following your suggestion, I have added 19 relevant papers and tried to improve the dialog with the existing literature in the literature review section, the methods section, and the discussion section. Thank you very much for your generous advice!

Round 2

Reviewer 2 Report

The contribution seems plausible.
The article has improved. 
Please detail more / explain this passage:

Among the three regions selected in this paper, Hong Kong and Macao are all typical two-level pyramidal structures, while Guangdong is a typical multi-hierarchical structure

Please detail more / explain this passage:

most of the de facto deep cooperative relationships are established between universities that are at the top of the pyramid of the three group levels,

Please detail more / explain this passage: 

The data source is the cooperation data generated by 35 universities in Guangdong-Hong Kong-Macao Great Bay Area jointly establishing 37 professional alliances which conducted 888 cooperation ties from 2017 to 2022.

The title needs correcting. Eliminate the “The” from the beginning. 
Should read:

Research on the Regional Cooperation Innovation Network of Universities in The Guangdong-Hong Kong-Macao Great Bay Area

Some minor errors eg should read “an organization” and “is unique”:

As a organization with the characteristics of both "professionalization" and "administrativeization", the cooperation behavior of universities has unique.

Author Response

Response to Reviewer 2 Comments (Round 2)

Point 1:  

Please detail more / explain this passage:

Among the three regions selected in this paper, Hong Kong and Macao are all typical two-level pyramidal structures, while Guangdong is a typical multi-hierarchical structure. most of the de facto deep cooperative relationships are established between universities that are at the top of the pyramid of the three group levels,

Response 1: This paragraph is based on the summary of the hierarchical analysis in 2.2, but as you pointed out, I did not fully explain the origin and meaning of this sentence, and I have explained it more fully in the new version, thanks for your comments!

Point 2:  

Please detail more / explain this passage:

The data source is the cooperation data generated by 35 universities in Guangdong-Hong Kong-Macao Great Bay Area jointly establishing 37 professional alliances which conducted 888 cooperation ties from 2017 to 2022.

Response 2: Based on your comments, I have reorganized the discussion at the beginning of 3.2 Data Source and Processing to clarify the acquisition of research data and the transposition process of the data matrix into two levels to make the meaning of the sentence clearer. 

Point 3:

The title needs correcting. Eliminate the “The” from the beginning. Should read:Research on the Regional Cooperation Innovation Network of Universities in The Guangdong-Hong Kong-Macao Great Bay Area

Response 3: Thank you for your generous correction, and I have made changes to the title as you suggested.

Point 4:

Some minor errors eg should read “an organization” and “is unique”:As a organization with the characteristics of both "professionalization" and "administrativeization", the cooperation behavior of universities has unique.

Response 4: I corrected the relevant grammatical errors according to your suggestion and submitted it to mdpi official for English editing and language correction.

Reviewer 3 Report

The authors responded with the utmost attention to the initial observations, considerably modifying the original article. I have no additional comments.

It is recommended to review the article from the point of view of the correctness of some phrases (for example, on lines 65-66 the phrase seems incomplete).

It is recommended to review again the article from the point of view of observing the rules regarding the citation guidelines.

Author Response

Response to Reviewer 3 Comments (Round 2)

Point 1:  

The authors responded with the utmost attention to the initial observations, considerably modifying the original article. I have no additional comments. 

Response 1: Thank you very much for your very specific and constructive comments on this article in the first round, which will help me to revise the article and improve my writing skills in the future. Thank you most sincerely!

Point 2:  

It is recommended to review the article from the point of view of the correctness of some phrases (for example, on lines 65-66 the phrase seems incomplete).

It is recommended to review again the article from the point of view of observing the rules regarding the citation guidelines.

Response 2: I corrected the phrase problem you mentioned and submitted the paper to mdpi's official language edit platform for a thorough check of the language and reference citation formatting rules.
